# Design and Experimental Demonstration of an Integrated Sensing and Communication System for Vital Sign Detection

**DOI:** 10.3390/s25123766

**Published:** 2025-06-16

**Authors:** Chi Zhang, Jinyuan Duan, Shuai Lu, Duojun Zhang, Murat Temiz, Yongwei Zhang, Zhaozong Meng

**Affiliations:** 1School of Transportation and Civil Engineering, Nantong University, Nantong 226019, China; zhangchi1845@stmail.ntu.edu.cn (C.Z.); 2233110231@stmail.ntu.edu.cn (J.D.); lushuai@stmail.ntu.edu.cn (S.L.); zdj@stmail.ntu.edu.cn (D.Z.); 2Department of Electronic and Electrical Engineering, University College London, London WC1E 6BT, UK; m.temiz@ucl.ac.uk; 3Department of Electronics and Electrical Engineering, Middle East Technical University, 06800 Ankara, Turkey; 4Department of Mechanical Engineering, Hebei University of Technology, Tianjin 300130, China

**Keywords:** integrated sensing and communications, micro-doppler, orthogonal frequency division multiplexing, software-defined radio, vital signs

## Abstract

The identification of vital signs is becoming increasingly important in various applications, including healthcare monitoring, security, smart homes, and locating entrapped persons after disastrous events, most of which are achieved using continuous-wave radars and ultra-wideband systems. Operating frequency and transmission power are important factors to consider when conducting earthquake search and rescue (SAR) operations in urban regions. Poor communication infrastructure can also impede SAR operations. This study proposes a method for vital sign detection using an integrated sensing and communication (ISAC) system where a unified orthogonal frequency division multiplexing (OFDM) signal was adopted, and it is capable of sensing life signs and carrying out communication simultaneously. An ISAC demonstration system based on software-defined radios (SDRs) was initiated to detect respiratory and heartbeat rates while maintaining communication capability in a typical office environment. The specially designed OFDM signals were transmitted, reflected from a human subject, received, and processed to estimate the micro-Doppler effect induced by the breathing and heartbeat of the human in the environment. According to the results, vital signs, including respiration and heartbeat rates, have been accurately detected by post-processing the reflected OFDM signals with a 1 MHz bandwidth, confirmed with conventional contact-based detection approaches. The potential of dual-function capability of OFDM signals for sensing purposes has been verified. The principle and method developed can be applied in wider ISAC systems for search and rescue purposes while maintaining communication links.

## 1. Introduction

Radio frequency (RF) sensing has become increasingly important in various applications such as defense applications, healthcare monitoring, security, smart homes, and search and rescue operations. Many approaches have been developed, such as ultra wide band (UWB), stepped frequency (SF), continuous wave (CW), and frequency-modulated continuous wave (FMCW) radars for various sensing purposes. Orthogonal frequency division multiplexing (OFDM) signals have also been considered as one of the promising waveforms for RF sensing due to their improved spectral efficiency and dual-function capabilities. Among them, vital sign detection has attracted the interest of many researchers since it plays a pivotal role in various fields, such as healthcare, security, and especially search and rescue operations [1,2].

Various microwave life detection systems and signal processing algorithms have been developed. Due to their superior range resolution, UWB radars are used to make indoor sensing in [3,4,5,6,7,8], and vital signs, including respiratory and heartbeat rates, have been detected. However, its short range, high path loss, and complex signal processing pose significant challenges. CW radars are more widely used as they are easy to implement but need a long data acquisition time, and no range information is available [9,10,11]. FMCW radars have been widely used in multiple applications for their characteristics, including localization, resilience to the noise generated by the surrounding environment, and capability of rejecting clutter and multipath [12,13,14]. On the other hand, OFDM, in addition to supporting communication, is potentially suited for sensing operations [15,16,17,18]. It is capable of dealing with multipath-caused interference, rejecting false targets, exhibiting better sensing capabilities, and having wider applicability in dynamic scenarios.

Micro-Doppler effects caused by respiratory chest movements (0.1–0.5 Hz, 0.1–1.2 cm displacement) and cardiac contractions (0.8–3 Hz, 0.01–0.1 cm displacement) are embedded in the propagation channel responses of communication systems [19]. Conventional sensing methods adopting OFDM technology carry out detection tasks with the channel state information (CSI) of the communication system. They are acquired by using least squares (LS) and linear minimum mean square error (LMMSE) estimators with pilot signals [20]. Both LS and LMMSE estimators involve discrete channel responses and statistics [21,22]. Hence, the frequency responses for sensing are effectively narrowband, as they are interleaved by the subcarriers for data transmission [23,24]. Zhou et al. employed the CSI of WiFi systems for indoor sensing, carrying out linear phase calibration (LPC) with robust principal component analysis (RPCA), and successfully demonstrated indoor sensing capability in a noisy environment [25].

This sensing requirement falls into the growing capability scope of integrated sensing and communication (ISAC) systems, where OFDM waveforms are commonly adopted to balance spectral and energy efficiency for communication and sensing purposes [26,27,28]. Experimental measurements of ISAC systems are necessary to validate their performance under various scenarios [29,30,31]. Human activity has been detected with an OFDM system implemented with GNU Radio [30]. Another study investigated vital sign detection using OFDM signals at 26 GHz [31]; however, such high carrier frequencies, e.g., 26 GHz, are not suitable for search and rescue operations since their penetration loss and path loss are significantly high. This study proposes a method to utilize OFDM signals for respiratory and heart rate detection in a typical indoor environment, and it is verified through experimental measurements via software-defined radio (SDR) platforms operating at 1.15 GHz carrier frequency. Our contributions include

The proposal of a method for vital sign detection using OFDM signals that can be utilized for communication simultaneously. It can be used to develop specific ISAC systems for healthcare applications or search and rescue systems, serving as a valid use case for ISAC systems.A three-tier data processing process for vital sign detection has been proposed to mitigate the noise effect and improve the precision of vital sign detection. Utilizing OFDM signals with 1 MHz bandwidth at 1.15 GHz RF carrier frequency for sensing makes the approach potentially deployable in various applications from smart healthcare to search and rescue operations.The experimental validation of the sensing function has been performed to verify the proposed method using SDRs; the merits of the OFDM-based ISAC system for the sensing function were highlighted in contrast to conventional radars.

The remainder of this paper is organized as follows: Section 2 details OFDM signal modeling and channel model. Section 3 describes the proposed method for vital sign detection. Section 4 presents experimental results on respiratory and cardiac signal detection, followed by discussion—Section 5 and conclusions—Section 6.

## 2. System Model

### 2.1. OFDM Signals

OFDM waveforms demonstrate superior flexibility for both sensing and communications compared to alternative dual-functional modulated signals [32], making them particularly suitable for ISAC applications. The complex baseband time-domain OFDM signal is formulated as [33](1)$$s\left(t\right)=\sum_{m=0}^{M-1}\sum_{k=0}^{N_{c}-1}X_{m,k}e^{j2\pi k\Delta f(t-mT_{\mathrm{sym}})}\cdot \mathrm{rect}\left(\frac{t-mT_{\mathrm{sym}}}{T_{s}}\right),$$
where *M* represents the total number of OFDM symbols in a time frame, $N_{c}$ denotes the total number of subcarriers, and $X_{m,k}\in C$ corresponds to the complex modulation symbol allocated to the *k*-th subcarrier of the *m*-th OFDM symbol. The subcarrier spacing and symbol duration are denoted by Δf and $T_{s}$, respectively, and $\Delta f=1/T_{s}$. Each symbol extends to $T_{\mathrm{sym}}=T_{s}+T_{\mathrm{cp}}$ through cyclic prefix insertion, where $T_{\mathrm{cp}}$ denotes the guard interval duration. The rectangular window function rect(τ) satisfies rect(τ)=1 for 0≤τ<1 and rect(τ)=0 otherwise, ensuring strict temporal confinement of individual OFDM symbols.

### 2.2. Sensing Channel Modeling

Figure 1 illustrates the measurement setup for heartbeat and respiration measurement using OFDM signals via SDRs, which are equipped with a single transmit antenna and a single receive antenna. The channel model of this setup can be modeled as a multi-path channel, where the propagation path is influenced by respiration and heartbeat, such that Doppler shifts are introduced to the reflected signal due to the respiration and heartbeat of the target. The multipath channel impulse response is given by [34](2)$$H\left(t,\tau \right)=\sum_{l=1}^{L}b_{l}\delta \left(t-\tau_{l}\right)e^{-j\left(2\pi f_{D,l}t+\phi_{l}\right)},$$
where *L* is the total number of multi-path components, with each path *l* having complex amplitude $b_{l}$ and propagation delay $\tau_{l}$, and δ(·) is the Dirac function. The Doppler shift $f_{D,l}=2v_{l}f_{c}/c$ results from target radial velocity $v_{l}$, where $f_{c}$ is the carrier frequency, *c* is the speed of light, and $\varphi_{l}$ is the phase variation of the *l*th path.

The chest movement-induced phase due to the heartbeat and respiration is modeled as [35](3)$$\phi \left(t\right)=\frac{4\pi d_{0}}{\lambda }+\frac{4\pi x(t)}{\lambda }-\varphi_{1}\left(t-\frac{2d_{0}}{c}\right),$$
where $d_{0}$ represents the nominal distance to the thorax and λ is wavelength. $x\left(t\right)=A_{r}\cos\left(\omega_{r}t\right)+A_{h}\cos\left(\omega_{h}t\right)$ models the time-varying chest displacement, where $A_{r}$ and $A_{h}$ represent amplitudes of respiratory and cardiac displacement, with the angular frequencies $\omega_{r}=2\pi f_{r}$ and $\omega_{h}=2\pi f_{h}$ corresponding to the rates of breathing and heartbeat. The term $\varphi_{1}\left(\cdot \right)$ accounts for phase perturbations during signal propagation, incorporating the effects of tissue heterogeneity and environmental factors. Assuming that the line-of-sight (LOS) path carries the heartbeat and respiration information due to the LOS measurement setup, including the micro-Doppler effects induced by the heartbeat and respiration, the modified channel model becomes(4)$$\begin{array}{cc} \; & H\left(t,\tau \right)=\;b_{1}\delta \left(t-\tau_{1}\right)\cdot \\ \; & \exp\left(-j2\pi f_{D,1}t-j\frac{4\pi }{\lambda }\left(d_{0}+x\left(t\right)\right)-j\varphi_{1}\left(t-\frac{2d_{0}}{c}\right)\right)\\ \; & +\sum_{l=2}^{L}b_{l}\delta \left(t-\tau_{l}\right)\exp\left(-j\left(2\pi f_{D,l}t-\phi_{l}\right)\right), \end{array}$$
where $\tau_{1}=2d_{0}/c$ represents the round-trip delay and $\phi_{l}\left(t\right)$ captures both static phase offset and time-varying perturbations of the *l*th path. The time-varying phase disturbance $\phi_{1}\left(t\right)$ introduces nonlinear distortions from environmental perturbations and tissue dielectric variations.

### 2.3. Time-Domain Received Signal

The baseband received signal incorporating both vital sign modulation and environmental multipath effects is formulated as(5)$$r\left(t\right)=\int_{-\infty }^{\infty }H\left(t,\tau \right)s\left(t-\tau \right)d\tau +n\left(t\right),$$
where H(t,τ) represents the time-varying channel impulse response defined in (Equation 4), s(t) denotes the transmitted OFDM waveform from Section 2.2, and $n\left(t\right)∼\mathrm{CN}\left(0,\sigma_{n}^{2}\right)$ denotes the complex additive white Gaussian noise (AWGN) with noise variance $\sigma_{n}^{2}$. Due to the first term of the channel model, corresponding to the LOS path, given by (Equation 4), the received signal r(t) contains the vital sign information through chest displacement modeled by $x\left(t\right)=A_{r}\cos\left(\omega_{r}t\right)+A_{h}\cos\left(\omega_{h}t\right)$.

### 2.4. Filtering Operation

The micro-Doppler shifts caused by displacement on the human body are nonstationary. The received signal from reflections is composed of broadband energy, and the instantaneous frequencies are widely distributed. The spectral analysis on the received signal is essential—isolating specific frequency harmonics associated with the subtle micro-Doppler effect requires filters with a high frequency definition. Low bandpass filters and very narrow frequency passbands are pivotal. Among the filter types designed to meet the demanding requirements of passband flatness, phase linearity, and stopband attenuation, Butterworth and Bessel filters were considered and proved to be valid for applications as such. A combination of them ensures frequency components can be separated with a high resolution.

**Butterworth Filter:** The Butterworth filter is characterized by its maximally flat magnitude response in the passband. Its mathematical formulation contains two key components. First, the squared magnitude response is defined as(6)$${\left|H\left(\omega \right)\right|}^{2}=\frac{1}{1+{\left(\frac{\omega }{\omega_{c}}\right)}^{2n}}$$
where $\omega_{c}$ represents the cutoff frequency in radians per second and *n* denotes the filter order. Second, the normalized low-pass transfer function can be expressed as(7)$$H\left(z\right)=\frac{1}{\prod_{k=1}^{n}\left(z-z_{k}\right)},\;\mathrm{with}\;\mathrm{poles}\;z_{k}=\omega_{c}e^{j\frac{(2k+n-1)\pi }{2n}}$$

This filter is fundamentally characterized by three distinctive attributes: a maximally flat passband magnitude response ensuring minimal amplitude distortion, a stopband attenuation rate proportional to filter order (−20n dB/decade), and a progressively nonlinear phase response. Precisely this unique combination of frequency domain characteristics—particularly a maximally flat passband and sharp stopband roll-off—makes the Butterworth filter favorably suitable for the fine micro-Doppler signature extraction discussed earlier. This unique combination of frequency domain characteristics renders the filter particularly advantageous in applications where precise spectral amplitude preservation constitutes a critical requirement, such as biomedical signal spectral analysis and radar micro-Doppler signature extraction. The spectrum of the filter for the two desired pass bands is shown in Figure 2.

**Bessel Filter:** The Bessel filter distinguishes itself through three intrinsic characteristics derived from its Bessel polynomial construction: a quasi-linear phase response ensuring minimal waveform distortion across the operational passband, an attenuation rate governed by filter order (−20n dB/decade) in the stopband, and group delay stabilization quantified by(8)$$\tau_{0}=\frac{(2n)!}{2^{n}n!\omega_{0}}\;(\mathrm{low}\text{-}\mathrm{frequency}\;\mathrm{approximation}).$$

These collective properties, particularly the temporal coherence preservation enabled by the frequency-independent group delay, established the filter’s preeminence in applications demanding strict temporal fidelity, especially for vital sign detection in ISAC systems where phase linearity crucially determines waveform integrity. For ISAC-based physiological monitoring, this temporal fidelity is essential to preserve micro-Doppler signatures of breathing and heartbeat in radar returns. The transfer function employing reverse Bessel polynomials is expressed as(9)$$H\left(z\right)=\frac{\theta_{n}\left(0\right)}{\theta_{n}\left(z/\omega_{0}\right)},\;\mathrm{where}\;\theta_{n}\left(z\right)=\sum_{k=0}^{n}\frac{(2n-k)!}{k!(n-k)!}z^{k}$$
with $\theta_{n}\left(z\right)$ representing the nth-order Bessel polynomial and $\omega_{0}$ denoting the reference frequency. The spectrum of the Bessel filter for the two desired pass bands is shown in Figure 3. The two distinct filters were applied to ensure that all the harmonics and intermodulation terms in the received signal by the nonlinear phase modulation are eliminated.

## 3. Data Acquisition and Detection Method

This section explains the signal processing steps, consisting of signal conditioning, pass-band filtering, and feature extraction, to determine the heartbeat and respiratory information from the received OFDM signals.

### 3.1. Data Processing

The received signals in the time domain were recorded as complex numbers. Given that the sampling rate was set to 1 MHz, this generates $2\times 10^{6}$ baseband samples (real and imaginary) per second. In the following section, we will use the first 30 s data from a 45-s recorded dataset to perform signal processing according to the steps given in Figure 4 to estimate the frequencies in corresponding to the respiratory and heart rates.

**Signal Conditioning:** We reshape the ($45\times 10^{6}\times 2$) samples into complex-valued signal data ($45\times 10^{6}$). Afterward, the first ($30\times 10^{6}$) complex-valued samples are chosen, and multistage anti-aliasing down-sampling is applied to reduce the sampling rate, lowering the original 1 MHz sampled OFDM signal to 100 Hz. The down-sampling equation is(10)y(n)=x(Mn),
where *M* is the factor for downsampling. After downsampling the original signal, the high-frequency components above the new Nyquist frequency ($f_{s}/M=50$ Hz) are filtered out, ensuring no interference from the power supply to the vital sign signals.

In the first stage, a finite impulse response (FIR) filter was adopted to remove high-frequency components and prevent aliasing, followed by decimation, reducing the signal sampling rate from 1 MHz to 10 kHz. At this stage, the complex signal is compressed to 1 × 300,000. In the second stage, the process of filtering and decimation is repeated: the signal sampling rate is reduced from 10 kHz to 100 Hz, further compressing the complex signal to 1 × 3000. This hierarchical downsampling improves computational efficiency while ensuring anti-aliasing. The bandwidth necessary for vital signs monitoring typically ranges from 0.1 to 5 Hz, and the 100 Hz here significantly reduces computational overhead while satisfying the Nyquist sampling theorem.

To correct the DC offset caused by the static background and hardware, we first subtract the mean value from the signal. Then we perform envelope detection on the processed complex signal to extract the amplitude values, resulting in a dataset having 1 × 3000 data points.

**Signal Filtering:** The downsampled data was still affected by various noises and interferences, which can distort the real signal characteristics and lead to inaccurate frequency estimates for respiration and heartbeat. Therefore, appropriate filtering is crucial. Initially, a fourth-order Butterworth filter was applied for preliminary bandpass filtering of the downsampled data, retaining the frequency range of 0.1 to 2 Hz for respiration and 1 to 3 Hz for heartbeat. The Butterworth filter has a flat passband response, which helps eliminate high-frequency noise interference; however, its phase response is nonlinear, necessitating further processing.

A Bessel filter, known for its nearly linear phase response, is employed to effectively preserve the signal characteristics. Using a fourth-order Bessel filter, the signal was further filtered within the bandpass frequency ranges of 0.1 to 0.5 Hz and 0.8 to 3 Hz, respectively. By cascading the Butterworth and Bessel filters, along with the zero-phase filtering technique described later, the high-fidelity extraction of the respiration and heartbeat information can be achieved. This method balances frequency band selection, phase linearity, and interference resistance. Next, bidirectional filtering is applied to the signal to eliminate group delays. This filtering technique is also known as zero-phase filtering, meaning that it does not introduce any phase distortion during filtering, thus preserving the phase characteristics of the signal.

Finally, since hardware and software filters cannot fully utilize preceding and succeeding data at the beginning and end of the signal, leading to inaccurate output, we apply boundary truncation to remove the first and last 5% samples of the data to eliminate transient effects and retain the valid signal.

**Feature Extraction:** To reduce spectral leakage and improve the accuracy and smoothness of spectral estimation when calculating the power spectrum, a Hamming window was applied to the data prior to performing the Fast Fourier Transform (FFT), with a 50% overlap to ensure that each segment of the signal was analyzed multiple times. The smooth decay characteristics of the Hamming window allow the signal to approach zero at the boundaries, thereby reducing the truncation effects. Additionally, its moderate main lobe width and significant side lobe attenuation enable the effective resolution of frequency components while suppressing noise. After applying the FFT, the power spectrum of the signal is obtained. Since the resulting spectrum is discrete, the frequency resolution is limited by the sampling duration ($△f=\frac{1}{T}$). The peak frequency of the actual signal may lie between two discrete frequency points, and directly taking the maximum index can introduce quantization errors. Therefore, a three-point parabolic interpolation method can be used to enhance the accuracy of the estimation.

Let the maximum index be *k* and the power spectrum values of three adjacent frequency points be y(k−1), y(k), and y(k+1). The corresponding frequencies are f(k−1), f(k), and f(k+1), with a frequency interval of Δf=f(k)−f(k−1). The quadratic fitting function is defined as(11)$$y=a\delta^{2}+b\delta +c$$
where δ denotes the offset relative to the midpoint f(k). The corrected peak frequency can be obtained by calculating δ.

The coefficients *a* and *b* are derived from the following formulas:(12)$$a=\frac{y(k-1)+y(k+1)-2y(k)}{2{\left(\Delta f\right)}^{2}}\;$$(13)$$b=\frac{y(k+1)-y(k-1)}{2\Delta f}$$

By differentiating the quadratic function and setting the derivative to zero, the offset δ at the vertex is calculated as(14)$$\delta =-\frac{b}{2a}=\frac{y(k-1)-y(k+1)}{2[y(k-1)+y(k+1)-2y(k)]}$$

The corrected peak frequency is then expressed as(15)$$f_{\mathrm{peak}}=f\left(k\right)+\delta \Delta f$$

The three-point parabolic interpolation method achieves sub-pixel-level frequency estimation by fitting three adjacent points near the spectral peak, thereby avoiding quantization errors introduced by directly selecting the maximum index. Algorithm 1 presents the entire signal processing algorithm explained and its steps.
**Algorithm 1** Vital Sign Detection—Multi-Physiological Parameter Analysis**Input:** Complex RF signal in binary format Rx_Data_TD**Output:** Respiratory frequency breath_freq and heart rate heart_freq**Step 1: System-Level Parameter Configuration**1: $Fs\_original\leftarrow 1\times 10^{6}$             ▹ Original sampling rate (1 MHz)2: Fs_new←100               ▹ Processing sampling rate (100 Hz)3: analysis_duration←60           ▹ Signal analysis duration (seconds)**Step 2: Signal Preprocessing**4: Open binary data file Rx_data_TD5: Read IQ quadrature components and reconstruct complex signal Rx_signal6: Multi-stage anti-aliasing downsampling: 1 MHz → 100 Hz7: Remove DC offset and extract signal amplitude feature real_signal**Step 3: Signal Filtering**8: Design bandpass filters to retain respiratory and heart rate frequency bands9: Apply zero-phase filtering to obtain real_breathsignal and real_heartsignal10: Eliminate boundary effects, retain middle 90% of valid data**Step 4: Feature Extraction**11: Use adaptive pwelch function to compute power spectral density px_breath and px_heart12: Use parabolic interpolation to estimate respiratory frequency breath_freq and heart rate heart_freq**Step 5: Output Results**13: Return respiratory frequency and heart rate

### 3.2. Simulation of the Method

Firstly, we checked and verified the feasibility of our detection method presented in Algorithm 1 in simulations. The simulation parameters are listed in Table 1.

Figure 5 demonstrates the signal processing outcomes of physiological monitoring using an OFDM-based ISAC system based on simulations. Data were generated according to the proposed system model. Figure 5a displays the time domain waveform after applying the 0.1–0.5 Hz bandpass filter, showing characteristic low-frequency oscillations with peak-to-peak amplitude variations between −1 and 1 during the 30-s observation window. Figure 5b presents the 0.8–2 Hz bandpass filtered signal in the time domain, revealing higher-frequency pulsatile components with steeper waveform transitions. The corresponding frequency-domain analyses in Figure 5c,d exhibit distinct spectral concentrations: the lower band (0.1–0.5 Hz) displays dominant spectral energy centered at 0.2 Hz, while the higher band (0.8–2 Hz) shows pronounced spectral peaks at 1.0 Hz, both aligned with the expected fundamental frequencies of respiratory and cardiac activities, respectively. The aforementioned simulation experiments systematically verified the effectiveness of the proposed algorithm in separating and characterizing different physiological signatures via frequency-selective processing. However, simulations are based on idealized conditions. To evaluate its practical performance, upcoming experiments will be conducted in real-world scenarios. Actual application environments are significantly more complex than the simulated environment.

## 4. Experiments and Results

This section introduces the experimental setup for vital sign detection and evaluates the results from the measurements. To experimentally validate the dual-functional performance of the ISAC system, we implemented a co-located transceiver configuration using two universal software radio peripheral (USRP) platforms B210 and X310 (Ettus Research, Austin, TX, USA). To validate the performance of the algorithm, ground truth physiological data—a respiratory pattern scientifically measured with the respiration monitoring belt unit and cardiac rhythms by electrocardiogram (ECG)—were recorded and compared with the results from the ISAC demonstrator.

### 4.1. Device Configuration

The system uses the USRP platform to transmit (B210 node) and receive (X310 node) OFDM signals. Two platforms were synchronized through a common clock from the laptop. OFDM frames were transmitted in a sensing cycle. The block diagram for the signal transmission and reception is shown in Figure 6, and the OFDM frame design for the ISAC demonstration system is shown in Figure 7. Due to throughput limitation of the network interface between the USRP and the host laptop, the instantaneous bandwidth of the OFDM signal was chosen to be 1 MHz, and this bandwidth, we believe, is sufficient to demonstrate the capability of the ISAC system once the vital signs can be successfully retrieved under the designated frequency bandwidth.

The signal for the OFDM-based ISAC system is configured to operate in the central frequency of 1.15 GHz with a total frequency bandwidth of 1 MHz, splitting into 64 subcarriers. At the beginning, the binary data stream was modulated with QPSK and then packaged into the OFDM frame. As illustrated in Figure 7, in each frame, the net data payload is 400 symbols, and another 48 symbols were added as headers for sequence control, filling up the specified space in the frame. The other spaces were allocated for pilots and the guard band between subcarriers.

The next step was to allocate subcarriers; 48 out of 64 subcarriers were used to carry data, and the other 16 were for overhead transmission. In each super frame, two synchronization OFDM symbols were added, forming the first two columns to facilitate synchronization and remove frequency offset in the receiving process. For the 64 subcarriers in each OFDM symbol, there were 4 for pilot signals, 1 for DC, and 48 for data, leaving 11 subcarriers unused. The complete frame forms a 64 × 12 matrix.

Before transmission at the antenna, the assembled frame in the frequency domain was converted into the time domain, and then CP (cyclic prefix) was added. The CP operation involves copying the last quarter of each OFDM symbol’s data in the time domain and appending it to the front. CP provides a guard interval, which helps to eliminate inter-symbol interference and mitigate multipath interference.

The setup of the synchronization sequence on the transmitter is crucial, and the core of timing synchronization lies in detecting the structure of the repeated training symbol corresponding to the two identical parts in the first half of Figure 7. The auto-correlation of the received signal r[n] over a sliding window is performed for time synchronization as [36](16)$$P\left[n\right]=\sum_{k=0}^{N/2-1}r\left[n+k\right]\cdot r^{*}\left[n+k+N/2\right]$$
where, *N* denotes the OFDM symbol length (excluding cyclic prefix). $r^{*}\left[n\right]$ is complex conjugate of r[n]. P[n] denotes the magnitude of the auto-correlation at time index *n*. By finding the perfect match between the training symbols, i.e., the highest auto-correlation, the time synchronization is completed.

In the receiver, USRP X310 equipped with an antenna is utilized to receive the signal. Upon receiving the radio frequency signals and obtaining the baseband data via RF chains, the first step is to convert it from serial to parallel format, resulting in an 80 × 12 matrix. Next, timing synchronization is performed, and then the frequency offset is compensated. Since we added a CP at the transmitter before the frame, it needs to be removed. After removing the CP, the frame structure becomes a 64 × 12 matrix corresponding to the time-domain frame. To extract the transmitted data stream, it is converted to the frequency via FFT with 64 points across 12 columns. This process yields the expected 64 × 12 frequency-domain frame structure. Over this data, symbol-based radar processing is performed to obtain the radar channel, hence extracting the target information. After that, the proposed method presented in Algorithm 1 is performed to obtain the heartbeat and respiratory information.

### 4.2. Scene Setup

Figure 8 illustrates the experimental configuration in a standard research office environment, where a host computer coordinates a B210 and X310 software-defined radio (SDR) platform (GNU radio 3.7.13.5) through USB 3.0 interfaces. The transmission chain employs a B210 unit connected via coaxial cable to a Matrix MG-RH02180 broadband antenna, while the X310 platform interfaces with an identical receive antenna through a low-loss coaxial link. The antenna array configuration maintained 20 cm element spacing at a 1 m elevation, forming a bistatic radar configuration with boresight alignment toward the standing human subject positioned 1 m from the antenna plane. We implemented an OFDM waveform generator and digital receiver processing chain through customized GNU Radio Companion (GRC) flowgraphs [37]. The version number of GRC is 3.7.13.5. The transmitted signal undergoes multipath propagation in the office environment, with target-reflected components captured by the receive antenna for vital sign monitoring. Baseband I/Q samples were recorded pre-equalization for subsequent respiratory and cardiac rhythm extraction through advanced signal processing algorithms. Concurrently, the adaptive equalizer outputs were monitored in real time through constellation diagram visualization (Figure 9) to ensure adequate communication link quality throughout data acquisition.

### 4.3. Experimental Results

Figure 10 illustrates the time domain signal extracted from the received data via the proposed method, where several long-period fluctuations can be observed, potentially corresponding to low-frequency respiratory signals. In addition, short-period spikes are visible on each waveform, which may represent high-frequency heartbeat signals. These fluctuations comprise the combined effects of respiration, heartbeat, and auxiliary harmonics, necessitating further filtering through specialized signal-processing techniques.

Figure 11 gives the comparison between the bandpass-filtered signal (0.1–0.5 Hz) and the reference signal from a respiration monitoring belt in the time domain. Both normalized waveforms (confined to [0, 1] amplitude over 30 s) exhibit nearly identical periodic oscillations reflecting respiratory cycles. The characteristic 8–9 cycles within the 30-s window correspond to 16–18 breaths per minute for both signals, remarkably aligning with the typical adult resting rate (12–20 breaths/min). Notably, the slight cycle irregularity—manifested through non-uniform waveform intervals—appears synchronously in both recordings, suggesting this variation likely originates from authentic physiological phenomena rather than measurement artifacts. This temporal and morphological coherence strongly supports the reliability of the extracted breathing pattern.

Figure 12 demonstrates synchronized periodicity between the bandpass-filtered signal (0.8–3 Hz) and the electrocardiogram (ECG)-derived reference signal in the time domain. Both normalized waveforms (15-s duration, amplitude normalized to [0, 1]) exhibit phase-aligned quasi-periodic fluctuations characteristic of cardiac rhythms. The filtered signal precisely mirrors the ECG’s cyclic pattern, with dominant oscillations recurring at 0.85-s intervals (71 bpm) for both signals—a rate consistent with healthy resting heart rates (60–100 bpm). Notably, the timing of systolic peaks shows millisecond-level alignment between the two signals, confirming their temporal coherence. While the amplitudes of the detected curves vary in magnitude (possibly due to motion interference), the core period is still strictly locked on the electrical activity of the ECG. This dynamic synchronization, spanning both frequency (heart rate matching) and phase (peak correspondence), validates the physiological origin of the extracted cardiac signal and its equivalence to gold-standard ECG monitoring.

Figure 13 depicts the spectrum of the lower bandpass signal (0.1–2 Hz). The red dashed line indicates the dominant peak obtained through parabolic fitting at 0.29 Hz, which we attribute to the respiratory process in the environment. This signal corresponds to a normal physiological breathing rate of 17.4 breaths per minute. Additionally, lower peaks occurred at approximately 0.6 Hz, 0.9 Hz, and 1.3 Hz, which may correspond to the second and third harmonics of respiration and the cardiac harmonic, respectively.

Figure 14 depicts the frequency domain characteristics of the cardiac signal within the high passband (0.8–3 Hz). The red dashed line indicates the dominant peak obtained through parabolic fitting at 1.27 Hz, which we attribute to the heartbeat process in the environment. This signal corresponds to a heart rate of 76.2 beats per minute. It is evident that the high passband effectively suppresses the noise introduced by low-frequency respiratory signals.

## 5. Discussion

Extensive studies demonstrate that OFDM signals can be utilized to detect vital signs, namely respiration and heartbeat rates, in addition to communication capacity. The scenario we adopted in this study was LOS for simplicity. Although search and rescue operations are generally performed under more complicated situations than LOS scenarios, and hence loss of the propagation paths will be more substantial and the channel will be more sophisticated, the proposed method is expected to still be valid when enough gain is compensated over propagation paths. These complex scenarios, including sensing through a wall or non-line-of-sight (NLOS) scenarios, will be investigated in future studies.

In the demonstration experiment, the processing time window is 30 s for the received signal, and vital signs have been successfully recovered through the obtained data. In contrast, in the simulation study, the duration of the generated data for processing is 10 s, which was sufficient for detecting vital signs. This information can be detected with the data of a shorter time window; however, the accuracy will decrease.

During the data acquisition period, there was only one person (the subject) in the room for the study. In order to avoid the disturbance caused by body movement at the beginning and end of measurement, only the data of 30 s in the middle of the recorded time period was used for processing. The presence of more subjects will be investigated in a future study.

The central frequency of the OFDM system in this study was 1.15 GHz, with a frequency bandwidth of 1 MHz. Although this frequency band has the ability to perform vital sign sensing and penetrate through certain walls if necessary, OFDM systems operating at a lower frequency band and with less bandwidth will be examined with the same approach to expand its application and improve reliability for more complex scenarios, such as search and rescue operations over rubble after earthquakes.

The OFDM-based ISAC system for vital sign sensing is compared with other state-of-the-art technology for similar aims, as shown in Table 2. The advantage of the ISAC approach is highlighted for its tradeoff capability among sensing and communication services, frequency bandwidth, and power. It exhibited its potential to be embedded with UAV for emergency search and rescue, where resources are limited in terms of power supply; weight carrying allowance; and signal processing capacity. Communication capability is markedly essential when the infrastructure is possibly damaged. The computational complexity in Table 2 only takes into account complex number calculations in the FFT operation for the designed algorithm.

The precision of the detection on a single subject based on the proposed method is given in Table 3. Among the trials in three human subjects, the worst case is 89% in accuracy, with the other detection tests giving a correct estimate. It is worth noting that the respiratory and heartbeat data from a respiratory monitoring belt and ECG do not have identifiable information; the radar and reference signals are one-dimensional electrical signals that are highly anonymous.

## 6. Conclusions

This study has proposed a method to detect vital signs using an OFDM-based ISAC system, validated through practical experiments. Vital signs, including breathing and heartbeat rates, have been successfully detected based on acquiring their micro-Doppler signatures impinged through reflection from the human body, and this was realized by adopting dual-functional OFDM signals in an LOS scenario and implemented on SDRs. Moreover, the communication performance is maintained when the data for sensing is acquired and recorded. By measuring the channel responses for closely spaced frequency subcarriers, the serious phase distortion in CSI-based detection techniques has been avoided effectively. The positive outcome consolidates the prospect of ISAC technologies in healthcare monitoring or search and rescue applications under more sophisticated situations, on top of its genuine communication capability. While this work can serve as a successful proof-of-concept ISAC demonstration for vital sign detection under controlled LOS conditions, more investigations on the impact of distance, multiple targets, and non-line-of-sight scenarios will be carried out to increase its capacity and reliability for practical implementations.

## Figures and Tables

**Figure 1 sensors-25-03766-f001:**
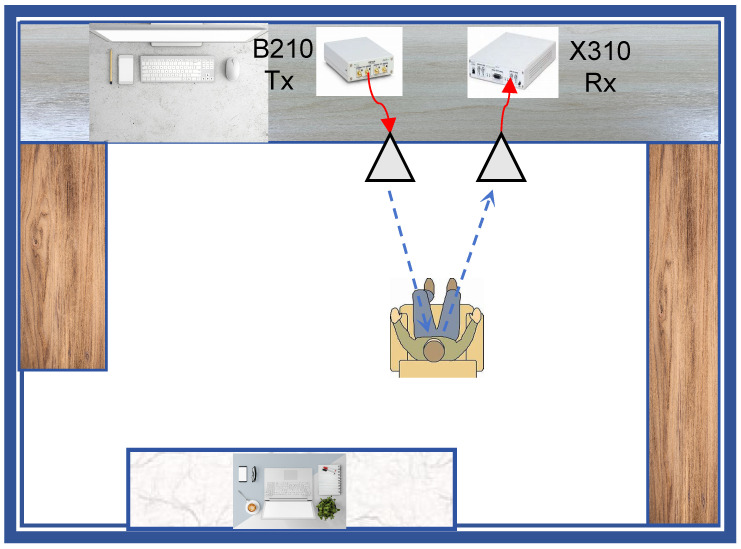
The conceptual setup of the vital sign detection system based on an ISAC system; B210 and X310 are collocated for Tx and Rx, respectively.

**Figure 2 sensors-25-03766-f002:**
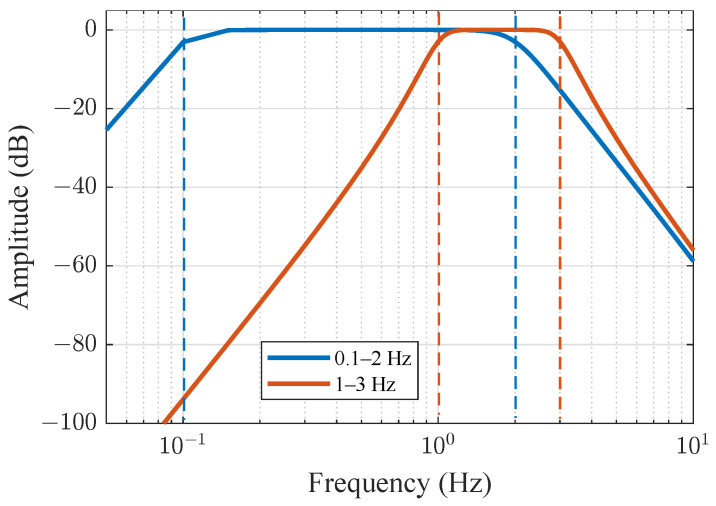
The spectra of the Butterworth filter for the 2 bandpass operations.

**Figure 3 sensors-25-03766-f003:**
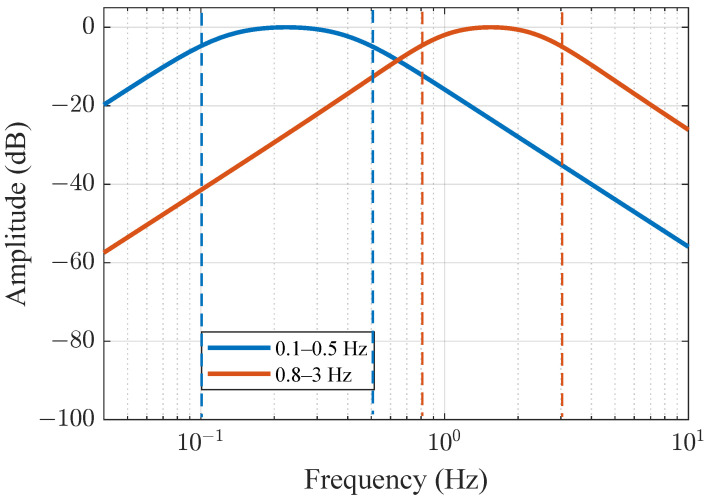
The spectra of the Bessel filter for the 2 bandpass operations.

**Figure 4 sensors-25-03766-f004:**
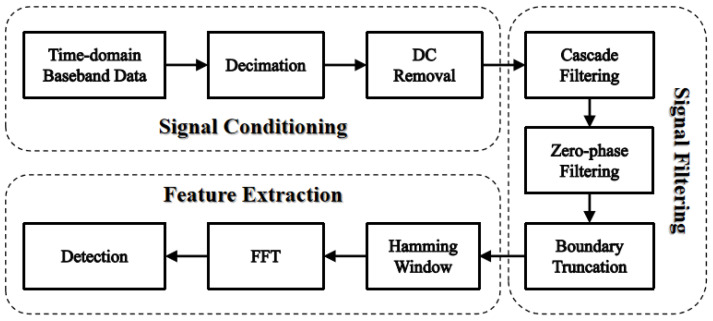
The flowchart of signal processing to detect the heartbeat and respiratory rates based on the received OFDM signals.

**Figure 5 sensors-25-03766-f005:**
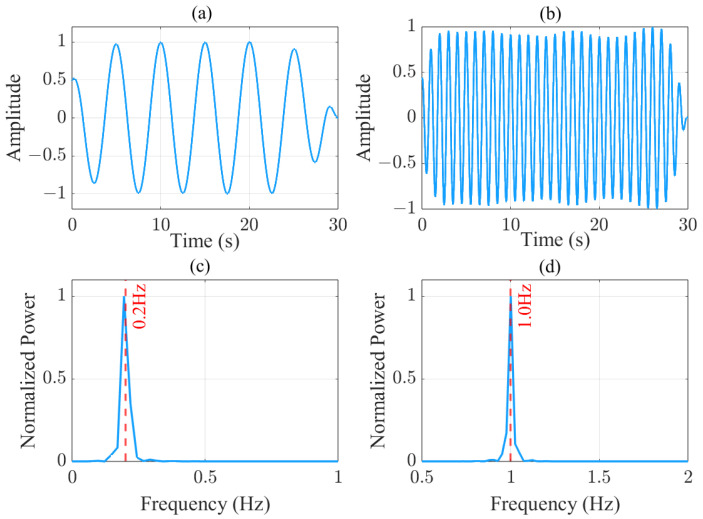
Simulation figures: (**a**) Time domain signal after 0.1–0.5 Hz pass-band filter. (**b**) Time domain signal after 0.8–3 Hz pass-band filter. (**c**) The frequency spectrum for the lower pass-band signal. (**d**) The frequency spectrum for the higher pass-band signal.

**Figure 6 sensors-25-03766-f006:**
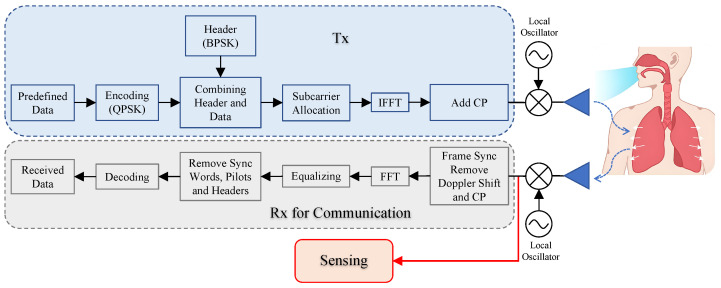
Schematic of the OFDM signal communication process in the ISAC demonstrator and data acquisition approach for sensing.

**Figure 7 sensors-25-03766-f007:**
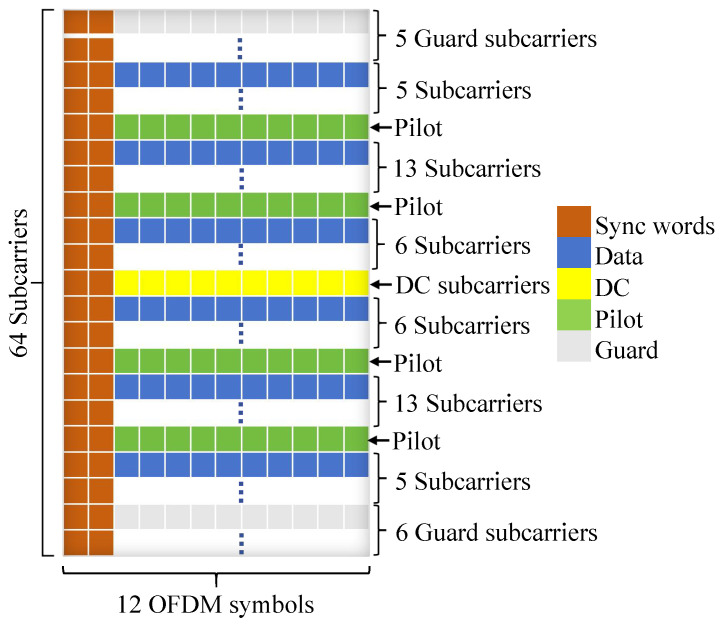
The OFDM signal frame design for the ISAC system.

**Figure 8 sensors-25-03766-f008:**
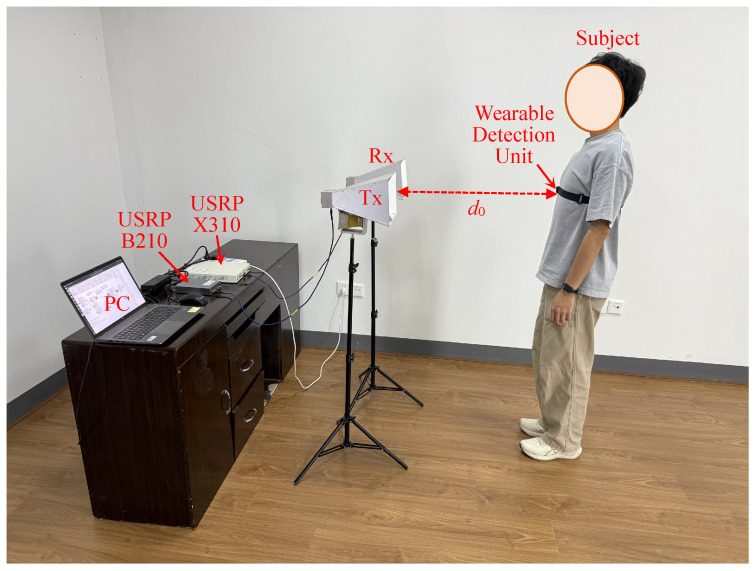
Experimental setup and measurements for vital sign detection using the ISAC system based on the USRPs.

**Figure 9 sensors-25-03766-f009:**
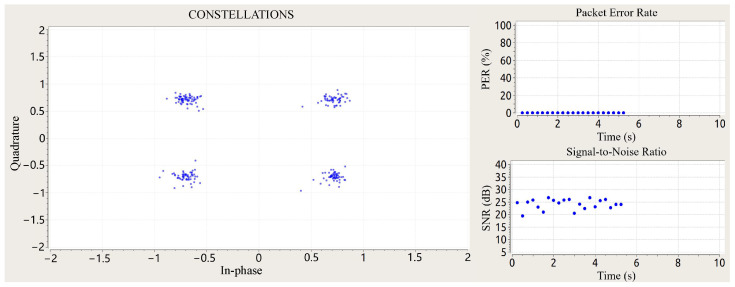
Received and demodulated OFDM communication signal and its constellation diagram (QPSK) while performing vital sign detection. The real-time signal demodulation performed by the communication receiver algorithm implemented on the SDR via GNU Radio software.

**Figure 10 sensors-25-03766-f010:**
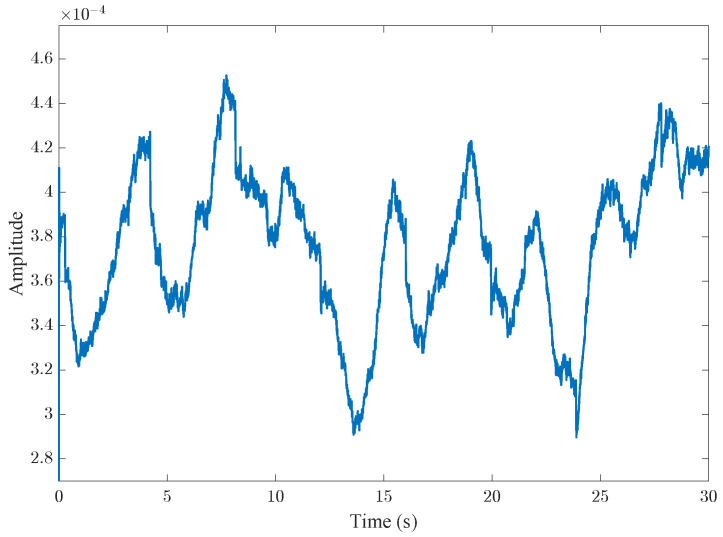
The received signal after down-sampling; the frequency for sampling of the signal became 100 Hz. One dominant harmonic is observed.

**Figure 11 sensors-25-03766-f011:**
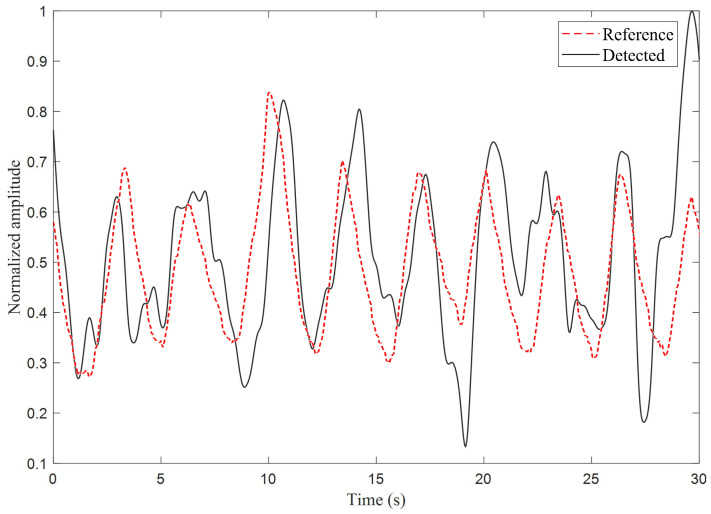
The bandpass-filtered signal in the time domain compared with the reference signal obtained from a respiration monitoring belt unit; the passband is between 0.1 Hz and 0.5 Hz.

**Figure 12 sensors-25-03766-f012:**
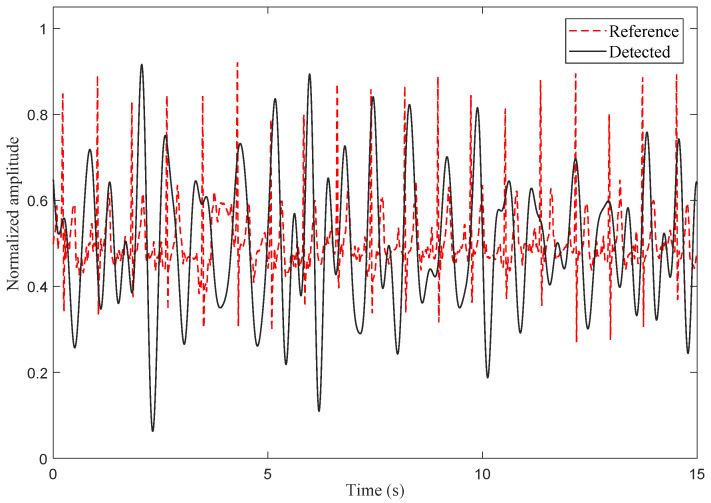
The down-sampled signal in the time domain after applying the bandpass filter; the pass band is between 0.8 Hz and 3 Hz, and the reference signal was based on the ECG.

**Figure 13 sensors-25-03766-f013:**
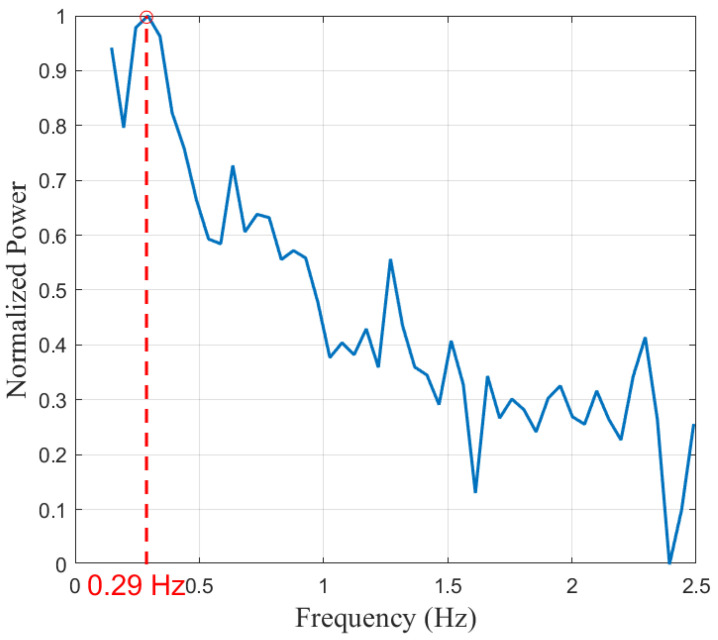
The frequency spectrum for the signal after applying the lower bandpass filter.

**Figure 14 sensors-25-03766-f014:**
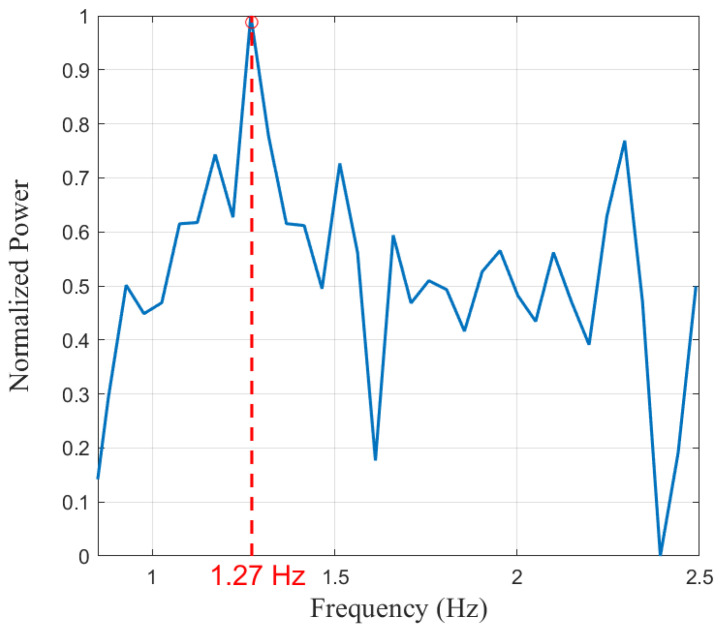
The frequency spectrum of the signal after applying the higher bandpass filter.

**Table 1 sensors-25-03766-t001:** Parameters of the OFDM-based ISAC system on SDR.

Parameters	Value	Description
$f_{c}$	1.15 GHz	Carrier frequency
▵f	15 KHz	Subcarrier interval
*N*	64	Number of subcarriers
$f_{s}$	1 MHz	Sample rate
*B*	1 MHz	Bandwith
$d_{0}$	1 m	Distance between antenna and subject
▵t	960 µs	TDD frame duration
$t_{sym}$	64 µs	Elementary symbol duration
$t_{cp}$	16 µs	Cyclic prefix duration
$P_{N0}$	−174 dBm/Hz	Noise power
$G_{Tx}$ $G_{Rx}$	60 dB	Transmitter and receiver gains
$G_{Antenna}$	7.0 dBi	Antenna gain
*P*	1	Number of transmit antenna
*Q*	1	Number of receive antenna
*O*	4	Filter order
$BF1_{\mathrm{low}}$	[0.1 2]	The Butterworth low bandpass filter
$BF1_{\mathrm{high}}$	[1 3]	The Butterworth high bandpass filter
$BF2_{\mathrm{low}}$	[0.1 0.5]	The Bessel low bandpass filter
$BF2_{\mathrm{high}}$	[0.8 3]	The Bessel high bandpass filter

**Table 2 sensors-25-03766-t002:** Comparison with the state-of-the-art contactless vital sign detection techniques.

Techniques	Central Frequency	Bandwidth	Time Window	Doppler Resolution	Transmit Power	Computational Complexity
CW [9]	450 MHz	N/A	25 s	0.0400 Hz	26 dBm	$(O\left(N\log N\right)∼O\left(N^{2})\right)$
	1.15 GHz	N/A	25 s	0.0400 Hz	24.8 dBm	$(O\left(N\log N\right)∼O\left(N^{2})\right)$
FMCW [38]	5.8 GHz	83.5 MHz	90 s	0.0111 Hz	13 dBm	$(O\left(N\log N\right)∼O\left(N^{2})\right)$
UWB [4]	4.3 GHz	1.7 GHz	120 s	0.0083 Hz	−9 dBm	$(O\left(N\log N\right)∼O\left(N^{2})\right)$
This work—ISAC	1.15 GHz	1 MHz	30 s	0.0333 Hz	10 dBm	$(O\left(M\log M\right)∼O\left(M^{3})\right)$

Variables: *N*: Number of samples, *M*: Number of subcarriers.

**Table 3 sensors-25-03766-t003:** Error evaluation for single-subject sensing.

Subject	Age	Respiration	Detected Respiration	Heartbeat	Detected Heartbeat
Person 1	24	0.29 Hz	0.29 Hz	1.27 Hz	1.27 Hz
Person 2	21	0.20 Hz	0.20 Hz	1.44 Hz	1.44 Hz
Person 3	21	0.17 Hz	0.15 Hz	1.17 Hz	1.19 Hz

## Data Availability

The data presented in this study are available on request from the corresponding author.

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
