# Peer review of "Design and Experimental Demonstration of an Integrated Sensing and Communication System for Vital Sign Detection"

_sensors, 2025, doi:10.3390/s25123766_

Round 1
Reviewer 1 Report
Comments and Suggestions for Authors
The research performed outstandingly in terms of scientificity and operability. The micro-Doppler detection of respiration and heartbeat within a certain frequency range under a specific carrier frequency was verified through SDR experiments. The error between the results and the reference equipment was less than 5%, and the equipment parameters, algorithm steps, and code framework provided the basis for reproduction. However, the experimental scene was limited to a single target within visual range and did not involve non-visual range or multi-target interference. Moreover, the influence of low-frequency bandwidth (1 MHz) on the wall-penetrating performance has not been quantified. The theoretical derivation of the multipath model has a weak correlation with the experimental results. The filter parameters have not been matched and analyzed with the measured spectrum. The universality demonstration and hierarchical depth are slightly insufficient.
(1) The current experiment has only verified the performance in a line-of-sight (LOS) single-target environment. Are there any plans to test the system in non-line-of-sight (NLOS) or multi-target scenarios in the future? How to deal with the challenges of multipath interference and target separation?
(2) In the text, Butterworth and Bessel filters are used for signal processing. How are the parameters, such as order and cut-off frequency, determined? Has the influence of different parameters on the accuracy of breathing/heartbeat detection been verified through simulation or experiments?
(3) In the experiment, the B210 and X310 devices need to keep their clocks synchronized. What specific calibration method should be adopted? Has the influence of clock deviation on micro-Doppler detection been quantified?
(4) Does the existing system have the ability to simultaneously detect multiple target vital signs? If expansion is needed, what improvements should be made in signal processing (such as spatial filtering) or hardware configuration (such as multi-antenna)?
(5) The research adopts a 1 MHz bandwidth. Has the signal attenuation of this configuration when penetrating common obstacles (such as brick walls, concrete) been tested? How to balance bandwidth and penetration capability to meet the needs of search and rescue?
(6) The current data processing requires a 30-second time window. Has the minimum detectable time window of the system been evaluated? What solutions exist at the algorithm or hardware level to shorten processing latency for real-time monitoring?
(7) The symbols of computational complexity mentioned in Table 2 lack mathematical derivations. Could you further explain their specific meanings? How can computing efficiency in actual hardware (such as SDR platforms) be optimized?
Reviewer 2 Report
Comments and Suggestions for Authors
The article demonstrates technical and methodological consistency; however, several adjustments are recommended to improve clarity, structure, and precision throughout the manuscript.
It would be beneficial to include a brief mention of the main results in the abstract to capture the reader’s interest and demonstrate the relevance of the work.
The manuscript uses acronym expansions inconsistently — some acronyms are defined, others are not. It is advisable to adopt a consistent rule throughout the text. For instance, on lines 24 and 26, the full text of the acronyms should be retained. The acronym OFDM is never written out in full, either in the abstract or in its first appearance in the main text. This should be corrected. Additionally, while UWB may be a familiar term for specialists, it is good practice to define all acronyms upon first use, including this one.
The phrase “such as healthcare, security, and in particular search and rescue operations” is repetitive and could be simplified to improve clarity and avoid redundancy.
On line 68, the phrase “This study proposed” is unnecessary and repetitive, since the contribution of the paper has already been introduced. It is recommended to remove this phrase.
On line 106, the notation used for the impulse function would benefit from including a period inside the parentheses for proper formatting and clarity.
The use of the symbol s as the input variable may be confusing, particularly because it is also used in Equation (7) to represent the complex frequency of the Laplace domain. This overlap should be clarified, or the variable name changed.
Although Section 2.4 introduces the specific filtering technique used in the article, there is no transition or explanation leading into it. As a result, the sudden presentation of Butterworth and Bessel filter theory appears disconnected. While including theoretical background makes the article more self-contained, it requires proper introduction to avoid abruptness.
On lines 148–149, the justification for using the Butterworth filter should be revised. It is recommended to state clearly that the filter is well-suited for micro-Doppler signature extraction. As written, it gives the impression that the topic is being introduced anew rather than developed from earlier discussion.
Likewise, on lines 158–159, the examples of Bessel filter applications could be rephrased to improve the clarity and tone of presentation.
On line 183, the expression “FIR filter downsampling” is technically imprecise. It gives the impression that downsampling is a function of the FIR filter itself. The sentence should be restructured to reflect that filtering is performed before downsampling, not as a single operation.
Between lines 191 and 193, the phrases “eliminate hardware offset” and “after removing the DC offset” are nearly synonymous and result in unnecessary repetition. The sentence also suggests that mean subtraction and DC removal are separate steps, which is misleading. Additionally, it is not clear whether envelope detection occurs before or after DC removal — this should be clarified.
On line 207, the text briefly mentions a dual-path filtering structure. A deeper explanation of this architecture would be valuable, though its inclusion remains at the discretion of the authors.
The methodology does not clearly specify the type of simulation performed. It is unclear whether the simulation was based on real-world data, a digital emulation using synthetic signals, or conducted in another environment. Details about the simulation setup should be included to support the claim on line 255 that a systematic analysis of the technological signature was conducted.
The inclusion of the proposed algorithm is commendable and adds value to the article.
However, the USRP device referenced on line 261 had not been introduced or described earlier in the manuscript. A brief explanation should be provided upon its first mention.
Figure 6 lacks sufficient explanation and connection to the proposed algorithm. It should be explicitly stated how this figure relates to the sensing path shown in Figure 4. In general, the figure and its caption can be rewritten to improve clarity.
While the article is understood to be a proof of concept, validating a method with only a single experimental instance is not ideal. Although adding more tests and uncertainty analysis may be beyond the scope of the current work, this limitation could be acknowledged and addressed in future research directions.
The reference signals from the belt sensor and ECG, presented in Section 4.3, are not introduced in the methodology. This is a gap that should be corrected for consistency and transparency.
On lines 361–362, there appears to be a mismatch: two descriptive labels are provided for three numerical values. The correspondence between descriptions and values should be clarified.
The methodological distinction between the demonstration experiment and the simulation is not explained in the methodology section, although it is referenced later in the discussion. This gap should be addressed to maintain coherence.
Finally, on page 19, the acronym LSTM is mentioned, but it is not actually used or explained in the text. If it is not necessary, it should be removed; otherwise, its use should be clearly justified and contextualized.
Reviewer 3 Report
Comments and Suggestions for Authors
This manuscript presents a method for detecting respiratory and cardiac pulsation signals from OFDM communication signals. The paper is well written overall and material is well presented. I have the following suggestions/questions for the authors:
- The final results are qualitatively presented for a single subject. It would be helpful to include more subjects so the accuracy of the method could be better evaluated by presenting metrics such as error in estimating heart rate or respiration rate.
- Some discussion around how this approach would work when the signal is reflected from multiple subjects would be helpful.
- Figures 13 and 14 can be combined for a more concise presentation.
- In figures 12, I see a random phase shift between the detected beat and the ECG R wave (the detected peak sometimes leads and sometimes lags the R wave). Similarly behaviour is present in figure 11. Adding some discussion around what could be causing this is useful.
- In figure 12, occasionally two peaks are detected for each R wave (for example around 6s and 13s). What could be causing this and how would this affect the accuracy?
- How is the performance of heart rate and respiration rate detection affected by distance of the subject from antenna?
- Figure 4 shows an IFFT step in the feature extraction method. Is this correct? The rest of the text seems to be referring to Welch power spectral analysis in this step.
